# Steroid Receptor Signallings as Targets for Resveratrol Actions in Breast and Prostate Cancer

**DOI:** 10.3390/ijms20051087

**Published:** 2019-03-03

**Authors:** Francesca De Amicis, Adele Chimento, Francesca Ida Montalto, Ivan Casaburi, Rosa Sirianni, Vincenzo Pezzi

**Affiliations:** Department of Pharmacy, Health and Nutritional Sciences University of Calabria, Via Pietro Bucci, Arcavacata di Rende, 87036 Cosenza, Italy; francesca.deamicis@unical.it (F.D.A.); adele.chimento@unical.it (A.C.); francescamontalto.93@libero.it (F.I.M.); ivan.casaburi@unical.it (I.C.); rosa.sirianni@unical.it (R.S.)

**Keywords:** Resveratrol, oestrogen receptor, androgen receptor, breast cancer, prostate cancer

## Abstract

Extensive research over the past 25 years in hormone-dependent cancers, such as breast cancer and prostate cancer, has identified the molecular mechanisms driven by steroid receptors, elucidating the interplay between genomic and non-genomic steroid receptors mechanism of action. Altogether, these mechanisms create the specific gene expression programs that contribute to endocrine therapy resistance and cancer progression. These findings, on the bidirectional molecular crosstalk between steroid and growth factor receptors pathways in endocrine resistance, suggest the use of multi-target inhibitors together with endocrine therapies, for treating resistant disease. In this review we will discuss the novel understanding on the chemopreventive and anti-cancer activities of Resveratrol (3,5,4′-trihydroxy-stilbene) (RSV), a phytoalexin found in grapes acting on a plethora of targets. We will highlight Resveratrol effect on steroid receptors signalling and its potential use in the treatment of hormone-dependent cancer. Understanding the molecular mechanisms by which the bioactive compound influences cancer cell behaviour, by interfering with steroid receptors functional activity, will help to advance the design of combination strategies to increase the rate of complete and durable clinical response in patients.

## 1. Introduction

Among steroid hormones, oestrogens and androgens, are known to influence many human hormone dependent cancers, such as those of the breast, prostate, ovary and endometrium among others, through different mechanism mediated by steroid receptors controlling cell proliferation and tumour development. In particular, breast and prostate cancer, share molecular similarities based on the idea that specific steroid receptors, the Oestrogen receptor α (ERα; referred to here as ER) for breast and Androgen receptor (AR) for prostate, have similar functions in driving both primary and recurrent disease. Clinical studies indicate the potential of steroid receptors as prognostic or diagnostic markers since alterations in these genes appear during tumour development and progression [1,2].

In recent years, research involving steroid receptors structure and cancer cell biology has resulted in opening up a unique window for novel targeted cancer therapies [3], aimed to the pharmacologic inhibition of signal transduction through steroid receptors, that prevents the binding of endogenous hormones. New molecules for cancer treatment must be validated in terms of their binding to steroid receptors, modification of their expression or alterations of their transcriptional activities [4]. Thus, steroid receptors are therapeutic targets for specific modulators designed to selectively influence their activity, for various pharmacological purposes [5].

Existing data indicate that signalling from ligand-activated steroid receptors leading to gene regulation are complex events that require a deeper knowledge. However, during the last couple of decades, progress has been made toward the comprehension of the steroid receptors’ activity in cancer cells. This has greatly helped the scientific community in designing more effective steroid receptor antagonists for the treatment of different types of cancer [6,7].

The inhibition of steroid receptors signalling, defined as endocrine treatment, is the therapeutic mainstay for patients with breast and prostate cancer [8,9,10]. Despite its efficacy, endocrine treatment fails for a proportion of patients with early disease who develop endocrine resistance, resulting in cancer progression. Specifically, tumour heterogeneity together with changes in ER and AR transcriptional activity, are responsible for endocrine therapy resistance. Thus, the design of new combination strategies for the treatment of these resistant tumours could increase the rate of complete and durable clinical response in patients. Recently, identification of the molecular mediators of endocrine resistance has been achieved, allowing the use of molecularly targeted agents to overcome or delay endocrine resistance. For instance, studies on the bidirectional molecular crosstalk between ER and growth factor receptor pathways in breast cancer patients [11] suggest the use of growth factor pathway inhibitors, rather than chemotherapy, for treating resistant disease. Besides, the non-therapy-specific and therapy-specific mechanisms of resistance in prostate cancer indicate that tumour cells display remarkable adaptability. The clinical diversity in disease course is a reflection of the marked molecular heterogeneity, demanding a personalized approach to the treatment of prostate cancer [12]. Thus, the challenge in treating resistant disease is to overcome the limited clinical benefits obtained by the current therapies. An effective treatment will require an agent that would act on multiple molecular targets exerting minimal toxicity.

In recent years, many naturally occurring compounds commonly present in the diet have gained considerable attention as antitumor agents [13,14]. In this regard, Resveratrol (3,5,4′-trihydroxy-stilbene) (RSV), a phytoalexin found in grapes, has been shown to exert preventive and antiproliferative effects against various types of cancer [15,16,17]. Numerous preclinical studies have demonstrated that RSV can directly inhibit proliferation and viability of cancer cells in a dose- and time-dependent manner. These events are related to its ability to induce growth inhibition, cell cycle arrest and apoptosis in several human cancer cell lines [15]. Additionally, RSV modulates ER- and AR-mediated transcription [18,19], although limited data support ER and AR as potential targets of RSV action.

In this review, we will discuss the evidences supporting a role for RSV as modulator of ER and AR in breast and prostate cancer, respectively.

## 2. ER Signalling in Breast Cancer

Numerous clinical and experimental data report that ER drives breast cancer development and progression by different mechanisms. ER genomic actions depend on ligand activated ER interaction with chromatin at the promoter region of target genes containing binding sites known as oestrogen responsive elements (EREs) [20,21]. Otherwise, the oestrogen-ER complex indirectly stimulates gene transcription through protein-protein interactions with activator protein 1 (AP1), specific protein 1 (Sp1), cyclic adenosine monophosphate (cAMP) response element binding protein (CREB) transcription factors. The full transcriptional regulatory activity is dependent on the recruitment of different transcriptional co-regulators, many of which have enzymatic activities able to modify chromatin structures and cooperate with the general transcriptional apparatus [22,23].

Oestrogen exerts also rapid, stimulatory effects on intracellular signal transductional pathways. Recent acquisitions demonstrate that, growth factors and signalling molecules from the tumour microenvironment may activate ER dependent non-genomic pathway, that begins outside the nucleus. This has clinical implications and suggests that targeting these pathways may provide opportunities for the treatment of ER+ breast cancer patients [24].

There has been an increasing interest on the post-translational modifications that further influence the stability, subcellular localization, trans-activity and hormone sensitivity of the ER. It is known that ER acetylation by p300, a histone acetyl-transferase (HAT), regulates ER trans-activity and hormone sensitivity [23,25]. Additionally, ER phosphorylation by mitogen-activated protein kinase (MAPK) and protein kinase A (PKA), enhances ER transcriptional activity [26]. Phosphorylation of Ser118 (S118) is one of the most well-characterized ER phosphorylation events. Recently, EGF-induced S118 phosphorylation has been suggested to increase breast cancer cell proliferation by activating a specific ER chromatin-binding profile. Importantly, the Cdk7 inhibitor, which inhibits S118 phosphorylation, suppresses growth of MCF7 breast cancer cells expressing either wild-type or mutant ER. Further, recent data on constitutively phosphorylated ER at S118 [27] highlight the importance of this phosphorylation in drug-resistant metastatic disease. PKA and p21-activated kinases 1 (Pak1) phosphorylate the hinge region of ER at S305 [28], making ER unresponsive to tamoxifen. This indicates that, even in the absence of oestrogen stimulation, growth factor signalling pathways could mediate ER activation altering its binding ability and, consequently, its transcriptional targets. These findings reveal potential avenues for pharmacological intervention in patients with mutated ER.

ER regulates expression of several genes involved in cell proliferation and cell cycle progression. The mitogenic effect of activated ER during G1-to-S transition relies on ER-mediated transcription of c-myc and cyclin D1 [29]. Besides, oestrogen-ER complex inhibits apoptosis by upregulating antiapoptotic Bcl-2 and Bcl-XL gene expression in breast cancer cells [30].

Recent acquisitions demonstrate that ER function is influenced by paracrine signalling from the tumour microenvironment contributing to breast cancer progression [31,32]. For example, fibroblasts co-culture with breast cancer cells has been shown to decrease ER gene expression in the cancer cells [33] and activate potent growth factor pathways for example, protein kinase B (AKT) and MAPK, thereby modulating the response of the epithelial cancer cells to anti-oestrogen treatment [31]. It has been discovered that the cytokines tumour necrosis factor α (TNFα) and interleukin 1β (IL-1β), produced by immune cells such as macrophages, can induce ER phosphorylation [34]. Cytokine-induced phosphorylation of ER results in ER binding to a subset of binding sites typically seen following oestradiol induction. Importantly, the cytokine-induced gene expression profile in MCF7 breast cancer cells can evoke metastatic events through ER but in a ligand independent manner. In contrast, TGFβ signalling by directly regulating ER function, blocks breast cancer progression; on the other hand, ER is able to inhibit TGFβ signalling [35]. A deeper understanding on how these pathways crosstalk, would be highly relevant for a better clinical translation.

## 3. Resveratrol Effects on ER Signalling: Potential Action in Breast Cancer Patients

A standard protocol for breast cancer treatment includes suppression of oestrogen production or action [36]. The first is achieved by blocking aromatase gene expression or enzymatic activities, the second by blocking ER binding to 17β-oestradiol (E2). During disease recurrence, antioestrogen treated tumours maintain ER expression, however signalling through the ER pathway is altered in these resistant tumours [11]. Several studies consider ER as a potential target of RSV, thus suggesting a role in the prevention of ER-dependent breast tumour and a possible therapeutic candidate for targeting ER in breast cancer [37].

In the presence of E2, RSV mitigates ER signalling, thereby generating an anti-proliferative expression signature that appears to be cell-type specific [38]. Particularly, RSV has been observed to reduce E2-stimulated cell growth and gene transcription in breast cancer cell lines. Based on the study by Basly et al. [39], the anti-estrogenic effects of RSV were demonstrated by transfection experiments in MCF-7 cells using a luciferase reporter containing ERE sites (See Figure 1 point 1). Other studies in different breast cancer cell lines indicate that RSV exerts mixed oestrogen agonistic/antagonistic activities in the absence of E2 but as an antioestrogen in the presence of E2 [40,41,42,43]. Specifically, RSV may act as a super-agonist at moderate concentrations (10–25 μM) activating hormone receptor mediated gene transcription. At low concentrations (0.1–1 μM) RSV acts as anti-oestrogen triggering pathways that inhibit oestrogen dependent effects involved in proliferation and transformation [39].

In this regard, our study [41] demonstrated that RSV reduces ER expression levels in MCF-7-TR breast cancer cells resistant to the ER antagonist 4-hydroxytamoxifen (4-OHT) through multiple mechanisms, including activation of p38 MAPK/casein kinase II (CK2) signalling, induction of p53 which binds, together with the transcription factor Nuclear transcription factor Y (NF-Y), the ER proximal promoter, inhibiting ER transcription (See Figure 1 point 2). Interestingly, we evidenced that different RSV concentrations (20, 50 and 100 µM) increased the number of cells within G0/G1phase of the cell cycle, with a concomitant decrease in the number of cells in S phase, suggesting a G1 arrest for both MCF-7 and MCF-7-TR cells. Additionally, the RSV dose used to inhibit MCF-7-TR cell proliferation was lower than the one used on parental cells [41].

Several studies prove the ability of RSV, in combinatorial therapy, to restore ER expression through an epigenetic mechanism [44,45]. For instance, RSV restores ER gene expression in ER-negative breast cancer cells by the enrichment of acetyl-H3, in particular of acetyl-H3-lysine9, and acetyl-H4, active chromatin markers in the ER promoter region [42]. Reactivation of ER expression by RSV was found to sensitize ER-negative breast cancer cells to the inhibitory effects exerted by 4-OHT on cell proliferation. E2 and 4-OHT further affected the ER-responsive genes in ER-reactivated MDA-MB-157 cells, suggesting an effective treatment option for hormonal refractory breast cancer [42].

RSV mechanism of action is strictly dependent on the molecular signature of the cancer model. RSV effects on xenografts or carcinogen-inducible in vivo models provide specific evidences. A recent interesting work investigated the effects of RSV on spontaneous mammary carcinogenesis using Delta-human epidermal growth factor receptor 2 (Δ16HER2) mice as HER2+/ER+ breast cancer model. Instead of inhibiting tumour growth, RSV treatment (daily intake of 4μg/mouse) reduced tumour latency and enhanced tumour multiplicity in Δ16HER2 mice [46]. This in vivo tumour-promoting effect of RSV was associated with down-regulation of ER protein levels and up-regulation of Δ16HER2 and its downstream pathway involving Target of Rapamycin Complex 1 (mTORC1)/Ribosomal protein S6 kinase (p70S6K)/ Eukaryotic translation initiation factor 4E-binding protein 1 (4EBP1), triggering cancer growth and proliferation [46].

Further study investigated the effects of a range of concentrations (0.5, 5, 50 mg/kg body weight) of RSV on mammary tumour development post-initiation, using immunocompromised mice [47]. The results demonstrate promotion of mammary tumour growth and metastasis by RSV in tumours derived from ER negative MDA-MB-231 and MDA-MB-435 cancer cell lines. Accordingly, the activity of Rac and PAK1, important mediators of cell invasion, measured in tumours from RSV treated mice was induced. Taken together, these findings implicate a potential tumour promoting mechanism of action exerted by RSV depending on the intrinsic molecular properties of the cancer model under investigation, such as ER positivity but also dependent on RSV concentrations. Thus, the beneficial effect of RSV in breast cancer patients should be re-evaluated, particularly in relation to risk groups that are susceptible to the development of ER-positive tumours.

It has been demonstrated that RSV inhibits E2-induced breast carcinogenesis via induction of nuclear factor erythroid 2-related factor 2 (NRF2)-mediated protective pathways. Female rats were treated with E2, RSV and RSV+E2 for 8 months [48]. RSV treatment induced NRF2-regulated antioxidant genes and inhibited E2-mediated alterations in NRF2 promoter methylation (See Figure 1 point 3). Moreover, RSV inhibited E2-mediated proliferative changes in mammary tissues, significantly increased tumour latency and reduced E2-induced breast tumour development [48].

RSV is able to inhibit non-genomic ER action controlling survival and proliferation in oestrogen responsive cells. Specifically, in MCF-7 cells, RSV increased the ER-associated Phosphatidylinositol-4,5-Bisphosphate 3-Kinase (PI3K) activity with a maximum stimulatory effect at concentrations as low as 10 μM; in contrast the higher concentration of 50 μM RSV diminished PI3K activity (See Figure 1 point 4) [49]. RSV action was ER-dependent since it could be blocked by the antioestrogen ICI 182,780. However, RSV induced the proteasome-dependent degradation of the ER [49].

Similarly, cell context influences the activation of non-genomic estrogenic pathways. For instance, in MCF-7 cells RSV alone did not significantly activate MAPK, PI3K or other non-genomic pathways, while repressed EGF-induced ERK activation [50]. These data are opposite to other reports indicating RSV ability, at the dose of 50 nM, to activate MAPK signalling in different ER+ cell types. Co-treatment with ER antagonists (ICI 182,780 or 4-OHT) blocked both RSV- and E2-induced MAPK and Endothelial nitric oxide synthase (eNOS) activation [51].

Further studies suggest that RSV is a novel candidate for prevention of tumour progression by inhibiting the cross-talk between ER and insulin-like growth factor receptor 1 (IGF-1R) signalling (See Figure 1 point 5]. RSV effectively reversed the BG-1 ovarian carcinoma cell proliferation induced by E2, down-regulating the expression of ER, cyclin D1, IGF-1R and activation of Insulin Receptor Substrate 1 (IRS-1) and AKT [52]. Then, RSV prevents the ER and IGF-1R cross-talk, inhibiting cell cycle progression. In the concern for novel therapeutic combinatory treatment, it needs to be mentioned that RSV functions as a cyclin dependent kinase (Cdk7) inhibitor, thus suppressing S118 ER phosphorylation [53].

A further important issue is that RSV may act as a pathway-selective ER ligand to modulate the inflammatory response but not cell proliferation. Particularly, RSV induces an altered coactivator-binding on the AF2 domain of ER, recruiting a group of co-regulators at the IL-6 promoter, suppressing IL-6 expression and consequently the inflammatory response in a variety of ER-positive tissues [54].

An elegant study examined the effects of RSV on carcinoma-associated fibroblasts (CAFs)-induced migration, invasion and self-renewal activity of breast cancer cells. RSV inhibited proliferation, migration and invasion of human breast cancer cells treated with CAF-conditioned media. These events were related to the suppression of the oestrogen-regulated gene cyclin D1 [55].

Aromatization of androgen precursors in adipose tissue is a major synthetic source of oestrogen after menopause, thus obesity is a risk factor for the development of hormone receptor-positive breast cancer driven by oestrogen. Recently, studies in mouse models and in women, report an obesity-inflammation-aromatase axis associated with NF-κB activation, elevated levels of pro-inflammatory mediators and increased aromatase expression. Using a mouse model of obesity, RSV was shown to suppress NF-κB binding activity, expression of pro-inflammatory mediators and aromatase in the mammary gland, thus reducing the estrogenic stimulus [56].

## 4. AR Signalling in Prostate Cancer

Prostate cancer is one of the most common forms of cancer, with more than 1 million new cases each year worldwide. AR signalling is central to the prostate cancer development, thus the primary treatment of prostate cancer is androgen deprivation by chemical or surgical castration [7]. However, a feature of prostate cancer is the high degree of tumour heterogeneity which is strictly related to resistance and progression to advanced androgen-resistant disease.

Clinical and experimental data widely suggest that AR signalling is the master regulator of prostate cancer growth and progression [57]. AR is located mostly in the cytoplasm where it interacts with heat shock proteins (HSPs), cytoskeletal proteins and other chaperones. Binding of its ligands, Testosterone or 5α-Dihydrotestosterone (DHT), induces a conformational modification resulting in nuclear targeting of AR and association with specific co-regulators [58]. Within the nucleus AR binds to tissue specific DNA sequences containing androgen responsive elements (AREs), facilitating the recruitment of HAT [59], co-regulators and the general transcription factors, thus activating transcription of androgen-dependent genes such as prostate specific antigen (PSA) and probasin [57]. The stability of AR-DNA and the amount of transcription initiated complexes are influenced by the ligands bound to AR. Receptor activity, at the genomic and non-genomic level, is influenced by the amount of AR protein regulated through proteasomal degradation [60]. Besides, proteolytic degradation is prevented by phosphorylation at serine residues, in androgen activated AR [61,62]. Indeed, AR is a substrate of receptor tyrosine kinases (RTKs), such as HER2, which can activate AR independently of androgens [63]. Moreover, MAPK family members extracellular regulated kinase (ERK), p38, c-Jun N-terminal kinase (JNK) and AKT enhance AR response to low levels of androgens and anti-androgens [64].

Newer therapies targeting AR are directed to the rapid non-genomic AR signalling [65,66]. Signalling molecules activated by AR and other steroid receptors in a non-genomic fashion include Src family kinases (SFK), Ras, MAPKs, AKT, protein kinase C (PKC), phospholipase C (PLC), epidermal growth factor receptor (EGFR) and other second messenger proteins [67,68,69]. The most well studied signalling molecule activated by AR is the membrane-associated non receptor tyrosine kinase Src, which plays an important role in lethal prostate cancer. The mechanisms that lead to aberrant Src signalling in tumours are not completely clear, however, dysregulated growth factors signalling leads to increased Src expression or activation. Src is found in an inactive form when associated with modulator of non-genomic action of oestrogen receptor (MNAR) [62]. AR recruitment to form a tertiary complex induces the activation of Src and its downstream effector, MAPK/ERK Kinase (MEK). This complex has a crucial activity in a castration-resistant LNCaP derivative prostate cancer cell line, suggesting that the new therapies acting on non-genomic AR signalling can find a potential clinical application [70]. MAPKs phosphorylation was activated following androgen stimulation of AR+ LNCaP cells or AR− PC3 cells stably expressing wild-type AR. This effect was abrogated in the presence of either non-steroidal anti-androgen bicalutamide or Src inhibitor PP1, demonstrating that MAPKs activation relies on the formation of the above mentioned complex. Besides, inhibition of Src in LNCaP cells decreased AR-dependent but androgen-independent cell proliferation induced by IL-8 [71]. Non-genomic AR signalling also promotes tumour cell survival through the activation of PI3K [72,73]. In PC3 cells stably expressing AR, proliferation and survival were stimulated by androgens which rapidly induced phosphorylation of AKT at Ser473, prevented by bicalutamide. Further reports demonstrated that non-genomic AR triggered Src-MAPKs-CREB activation [74]. In prostate cancer models, inhibition of Src decreased prostate cancer cell adhesion, migration and invasion, while targeting both Src and Abl (Abelson murine leukaemia viral oncogene homolog) attenuated lymph node metastasis of orthotopic PC3 xenografts [75]. In fact, it was demonstrated the ability of AR to enhance invasion of prostate cancer cell lines, independently of its nuclear localization but in a Src-dependent manner [76].

Recent acquisitions strongly indicate that reciprocal tumour-stroma interactions contribute to AR signalling in prostate cancer [77]. Particularly, changes in the tumour adjacent stroma lead to aberrant cancer growth, influencing the recruitment of infiltrating inflammatory cells. The release of specific mediators creates a *milieu* that favours a cross-talk between tumour cells and the adjacent microenvironment generating a persistent inflammatory response. The reactive stroma consists largely of CAFs which could contribute either positively or negatively to tumour growth. Experimental models support the idea that AR mediates, in part, the complex molecular interaction between CAFs and prostate cancer. Altered AR target genes in CAFs could affect AR-mediated regulation of growth, adhesion, motility and invasion of prostate cancer cells [78]. AR function in prostate CAFs was shown to be regulated by hydrogen peroxide-inducible gene 5 (Hic-5), an AR co-regulator, involved in the regulation of a limited number of genes activated via both genomic and non-genomic AR-dependent mechanisms [79]. Hic-5 acts also in the cytoplasm of CAFs preventing active fibroblast guidance of cancer cell movement and metastasis [80].

## 5. Resveratrol Effects on AR Signalling: Potential Action in Prostate Cancer Patients

Prostate cancer heterogeneity represents a challenge for clinical interventions. Thanks to its ability to reduce AR expression [19] and inhibit multiple targets [81], RSV may represent an ideal therapeutic compound. The inhibitory effects of RSV on androgen action in AR+ prostate cancer cells are well documented. RSV represses AR at the protein or mRNA level and, consequently, different classes of androgen up-regulated genes, including PSA, human glandular kallikrein-2, AR-specific coactivator ARA70 and the cyclin-dependent kinase inhibitor p21 [82,83].

RSV inhibits the androgen receptor signalling in tumour prostatic cells, inhibiting cell proliferation. This effect is achieved by both decreasing the production of AR agonists, since recent studies, have shown that RSV treatment, inhibits CYP17A1 in adrenocortical cells [84] (See Figure 2 point 1) and by blocking the receptor activity. Gao et al. [83] found that RSV effects on AR activity are concentration dependent; AR activity is enhanced at low concentrations while repressed at higher. Harada et al. [85] recently reported that RSV represses AR targets gene expression (See Figure 2 points 2,3), at least partially, by enhancing AR degradation in a time- and dose-dependent manner.

Further evidences obtained by DNA microarray analysis of the transcriptional program induced by RSV treatment of LNCaP cells, reveal an early and sustained decrease in the expression of many androgen-responsive genes that does not parallel AR decrease at the protein level. RSV did not oppose all transcriptional changes induced by androgens [86] indicating that RSV protection against prostate cancer is not strictly attributable to repression of AR expression. Thus, RSV reduces the AR protein levels but the reduction could not totally explain the suppression of AR function, as demonstrated in prostate cancer cell lines. Particularly, total and nuclear AR levels were not affected after incubation, whereas RSV inhibited the binding of AR to the enhancer region of PSA and decreased the acetylation of AR even at an early phase (See Figure 2 point 4). At a later phase after incubation, the ligand-induced nuclear accumulation of AR was markedly decreased by RSV. Therefore, RSV prevents DNA binding of AR, probably by reducing its acetylation status [87].

Transition of prostate cancer to the castration-resistant phenotype correlates with accumulation of a splice isoform of AR called AR-V7, that acts as a constitutively active transcription factor. Patients become refractory to conventional therapy because of the activity of this AR isoform. Interestingly, RSV is able of inhibiting AR-V7 transcriptional activity by downregulating AR-V7 protein levels in ectopically expressed AR-V7 PC3 cells, an AR-negative prostate cancer cell line [88]. Of note, RSV does not affect the mRNA levels of AR-V7 nor its nuclear translocation, instead enhances AR-V7 poly-ubiquitination and subsequent proteasome-mediated degradation. These findings suggest that RSV could be a potential treatment for AR-V7-positive castration resistant prostate cancer [88].

Further studies in prostatic cell lines showed that RSV repressed transcriptional activity of the AR through c-Jun specific DNA binding activity (See Figure 2 point 3). For instance, c-Jun as well as its phosphorylated form increased dose-dependently after RSV treatment. Overexpressed c-Jun mediates, then, an inhibitory effect on the function of AR, further supporting the idea that this polyphenol might potentially be useful in prostate cancer treatment [89].

Given the prominent role of PI3K/ Phosphatase and tensin homolog (PTEN) pathway in prostate cancer, RSV effects on this pathway have been evaluated. RSV inhibited PI3K activity by inhibiting AR action and by stimulating PTEN expression [90] (See Figure 2 point 5), Additionally, PI3K activity in AR-/ER+ PC-3 prostate cancer cells can be modulated by RSV via inhibition of ER [91].

Although most of the knowledge on RSV and prostate cancer comes from in vitro studies, there are, however, fewer papers that explored RSV action and cancer outcomes *in vivo*. An elegant study was conducted on a transgenic adenocarcinoma mouse model of prostate cancer fed RSV (625 mg/kg) or phytoestrogen-free control diet, starting at 5 weeks of age. RSV significantly reduced the incidence of poorly differentiated prostatic adenocarcinoma by 7.7-fold [92]. In the dorsolateral prostate, RSV significantly inhibited cell proliferation, increased androgen receptor and IGFIR and significantly decreased IGF-1 and ERK1 phosphorylation (See Figure 2 point 6), demonstrating a biochemical mechanism that warrants for RSV-mediated inhibition of prostate cancer development in vivo [92]. Additionally, dietary RSV lowered the incidence of prostatic adenocarcinoma in the transgenic adenocarcinoma mouse prostate (TRAP) model via down-regulating AR expression and mRNA levels of androgen-responsive glandular kallikrein 11, which has been determined to be an ortholog of the human PSA gene [93].

## 6. Conclusions

Breast and prostate cancer are part of the ongoing therapeutic revolution currently underway in the medical oncology, although the perspective to reduce disease progression still remains a significant challenge. Data reported in this review indicate that breast and prostate cancer can be responsive to RSV, which will affect multiple targets. RSV possesses a plethora of biological and chemopreventive activities targeting ER and AR, therefore it holds great promise for future development of optimal combination treatment regimens. RSV acts on molecular pathways that become constitutively activated in endocrine-resistant tumours, thus it might become part of future therapeutic protocols. Further studies are needed to evaluate benefits of combination therapy, particularly on early diseases. Hopefully, results of these studies will contribute to further shape the future of breast and prostate cancer therapy.

## Figures and Tables

**Figure 1 ijms-20-01087-f001:**
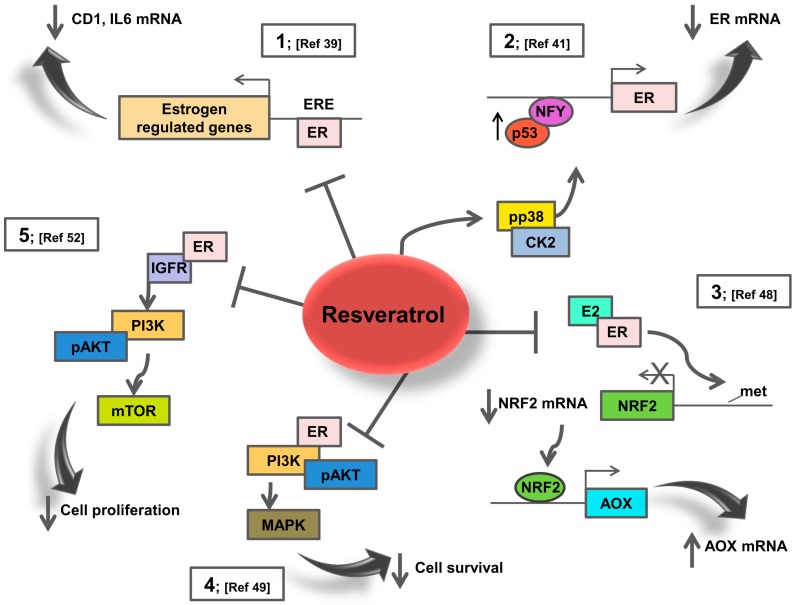
Summary of Resveratrol effects on ER dependent pathways. 1. RSV represses different classes of oestrogen up-regulated genes in breast cancer cells. 2. RSV activates p38 MAPK/CK2 signalling, induces the recruitment of a p53/NFY complex at the ERα proximal promoter reducing ER mRNA and protein levels in tamoxifen-resistant breast cancer cells. 3. RSV induces NRF2-regulated antioxidant genes and inhibits E2-mediated alterations in NRF2 promoter methylation in mammary cells. 4. RSV inhibits the ER-dependent PI3K activity in MCF-7 breast cancer cells. 5. RSV inhibits the cross-talk between ER and IGF-1R signalling in ovarian cancer cells.

**Figure 2 ijms-20-01087-f002:**
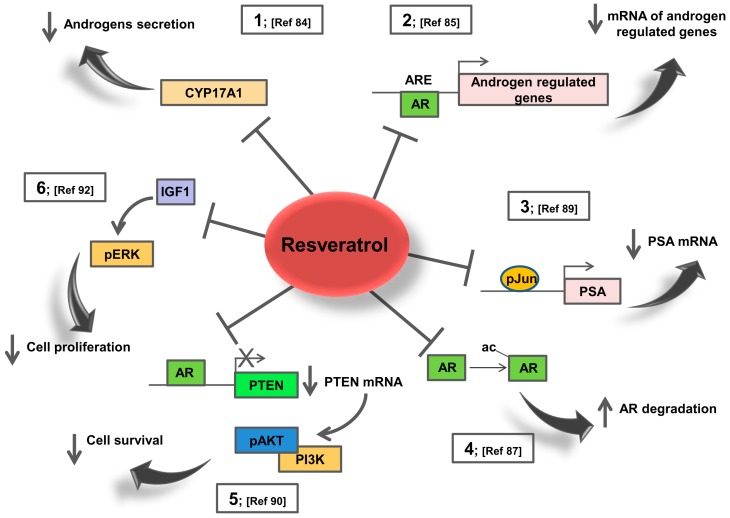
Summary of Resveratrol effects on AR dependent pathways. 1. RSV reduces the production of adrenal androgens, via inhibition of CYP17A1 in adrenocortical cells. 2. RSV represses different classes of androgen up-regulated genes in prostate cancer cells. 3. RSV decreases the transcriptional activity of AR through c-Jun specific DNA binding activity in prostatic cancer cells. 4. RSV decreases the ligand-induced nuclear accumulation of AR reducing its acetylation status in prostate cancer cells. 5. RSV inhibits PI3K activity by stimulating PTEN expression and decreasing AR action in LNCaP cells. 6. RSV decreases IGF-1 and ERK1 phosphorylation in transgenic adenocarcinoma mouse model of prostate cancer.

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
