# Peer review of "Steroid Receptor Signallings as Targets for Resveratrol Actions in Breast and Prostate Cancer"

_ijms, 2019, doi:10.3390/ijms20051087_

Round 1

Reviewer 1 Report

This is a comprehensive review on the anti-tumor activities of resveratrol against breast and prostate cancers through targeting estrogen or androgen receptor signaling pathways. In the review the authors provide detailed background information on the important roles that the steroid receptors and their signaling pathways play in hormone-dependent cancers. Following the introduction, the anti-tumor properties of resveratrol against breast and prostate cancers were summarized, with specific focus on the compound’s effect against ER/AR signaling pathways. Overall the review highlights the therapeutic potential of resveratrol against breast and prostate cancers through targeting steroid receptor signaling pathways.

Suggestions:

1. Only breast and prostate cancers were focused in the review. It’s better to point out this in the title.

2. A very interesting tumor promoting activity was mentioned at line 180-189 (reference 37). Are there any related/similar studies reporting similar outcomes? Should this be a concern regarding the safety of resveratrol in cancer treatment? Some discussion on this can be added.

3. Although studies contributing to Figures 1 and 2 were specifically referred, it’s better to also include their reference numbers in the figure captions.

Author Response

Answers to Reviewers

We are grateful for the reviewers’ comments

Reviewer #1

1. Only breast and prostate cancers were focused in the review. It’s better to point out this in the title.

Thank you for your suggestion. In the new version, the title has been modified as follows: “Steroid receptor signallings as targets for Resveratrol actions in breast and prostate cancer”

2. A very interesting tumor promoting activity was mentioned at line 180-189 (reference 37). Are there any related/similar studies reporting similar outcomes? Should this be a concern regarding the safety of resveratrol in cancer treatment? Some discussion on this can be added.

Thank you. In agreement with the reviewer request, in the revised version of the manuscript, at the line 193, we provide a further study regarding the tumor promoting activity of Resveratrol.

“Further study investigated the effects of a range of concentrations (0.5, 5, 50 mg/kg body weight) of RSV on mammary tumor development post-initiation, using immunocompromised mice (Castillo-Pichardo, L.; Cubano, L. A.; Dharmawardhane, S., Dietary grape polyphenol resveratrol increases mammary tumor growth and metastasis in immunocompromised mice. BMC complementary and alternative medicine 2013, 13: 6, doi: 10.1186/1472-6882-13-6). The results demonstrate promotion of mammary tumor growth and metastasis by RSV in tumors derived from ER negative MDA-MB-231 and MDA-MB-435 cancer cell lines. Accordingly, the activity of Rac and PAK1, important mediators of cell invasion, measured in tumors from RSV treated mice was induced. Taken together, these findings implicate a potential tumor promoting mechanism of action exerted by RSV depending on the intrinsic molecular properties of the cancer model under investigation, such as ER positivity, but also dependent by RSV concentrations. Thus the beneficial effect of RSV in breast cancer patients should be reevaluated, particularly in relation to risk groups that are susceptible to the development of ER-positive tumors”.

3. Although studies contributing to Figures 1 and 2 were specifically referred, it’s better to also include their reference numbers in the figure captions.

 In the new version, in the Figure 1 and 2 we included the specific references.

Reviewer 2 Report

Major

1.Please provide more updated references especially after 2016, most of the references are older than 2010.

Minor

1.Line 41 “Androgen receptor” should be “androgen receptor” 

2.Line 149-152 “Specifically, RSV may act as a super-agonist at moderate concentrations (10-25 mM) activating hormone receptor mediated gene transcription. At low concentrations (0.1-1 mM) RSV acts as anti-estrogen triggering pathways that inhibit estrogen dependent effects involved in proliferation and transformation.” Need to add with references.

3.Line 173 “Several studies prove…” this sentence should follow with more than one reference.

4.Please revise the format of every “et al” as “et al.”

5.Line 310 “dependent manner-” should be “dependent manner.”.

Author Response

Reviewer #2

1.Please provide more updated references especially after 2016, most of the references are older than 2010.

As requested by the reviewer, in the new version, we updated references (identified in the text by yellow highlight) especially after 2016, as reported below:

The references 1 and 2 are deleted and replaced, respectively, with the following new references:

·         Louie MC and Sevigny MB. Steroid hormone receptors as prognostic markers in breast cancerAm J Cancer Res. 2017; 7(8): 1617–1636.

·         Fujita K, Nonomura N. Role of Androgen Receptor in Prostate Cancer: A Review. World J Mens Health. 2018 Sep 10. doi: 10.5534/wjmh.180040.

The references 3 and 4 are deleted and replaced with the following new reference:

·         Groner, A C.; Brown, M., Role of steroid receptor and coregulator mutations in hormone-dependent cancers. Journal of Clinical Investigation 2017, 127, (4), 1126-1135.

At the line 51, we added the new reference:

·         Ulm M, Ramesh AV, McNamara KM, Ponnusamy S, Sasano H, Narayanan R. Therapeutic advances in hormone-dependent cancers: focus on prostate, breast and ovarian cancers. Endocr Connect. 2019 Feb 1;8(2):R10-R26. doi: 10.1530/EC-18-0425.).

The reference 5 is deleted and replaced with the following new reference:

·         Simons, S. S.; Edwards, D. P.; Kumar, R., Minireview: Dynamic Structures of Nuclear Hormone Receptors: New Promises and Challenges. Molecular Endocrinology 2014, 28, (2), 173-182.

At the line 57, we added the new references:

·         Sharma, D.; Kumar, S.; Narasimhan, B., Estrogen alpha receptor antagonists for the treatment of breast cancer: a review. Chem Cent J 2018, 12.

·         Gucalp, A.; Traina, T. A., The Androgen Receptor: Is It a Promising Target? Ann Surg Oncol 2017, 24, (10), 2876-2880.

At the line 59, we added the new references:

·         Nevedomskaya, E.; Baumgart, S. J.; Haendler, B., Recent Advances in Prostate Cancer Treatment and Drug Discovery. International journal of molecular sciences 2018, 19, (5).

·         Tong, C. W. S.; Wu, M. X.; Cho, W. C. S.; To, K. K. W., Recent Advances in the Treatment of Breast Cancer. Front Oncol 2018, 8.

The reference 7 is deleted and replaced with the following new reference:

·         Lousberg, L.; Collignon, J.; Jerusalem, G., Resistance to therapy in estrogen receptor positive and human epidermal growth factor 2 positive breast cancers: progress with latest therapeutic strategies. Ther Adv Med Oncol 2016, 8, (6), 429-449.

The reference 8 is deleted and replaced with the following new reference:

·         Ciccarese, C.; Massari, F.; Iacovelli, R.; Fiorentino, M.; Montironi, R.; Di Nunno, V.; Giunchi, F.; Brunelli, M.; Tortora, G., Prostate cancer heterogeneity: Discovering novel molecular targets for therapy. Cancer treatment reviews 2017, 54, 68-73.

The references 9 is deleted and replaced with the following new references:

·         Thakur, R. S.; Ahirwar, B., Natural Compounds A Weapon to Ameliorate Breast Cancer Cells: A Review. Anti-Cancer Agent Me 2017, 17, (3), 374-384.

·         Bonofiglio, D.; Giordano, C.; De Amicis, F.; Lanzino, M.; Ando, S., Natural Products as Promising Antitumoral Agents in Breast Cancer: Mechanisms of Action and Molecular Targets. Mini-Rev Med Chem 2016, 16, (8), 596-604.

The references 10 is deleted and replaced, respectively, with the following new references:

·         Rauf, A.; Imran, M.; Butt, M. S.; Nadeem, M.; Peters, D. G.; Mubarak, M. S., Resveratrol as an anti-cancer agent: A review. Crit Rev Food Sci 2018, 58, (9), 1428-1447.

·         Conte, A.; Kisslinger, A.; Procaccini, C.; Paladino, S.; Oliviero, O.; de Amicis, F.; Faicchia, D.; Fasano, D.; Caputo, M.; Matarese, G.; Pierantoni, G. M.; Tramontano, D., Convergent Effects of Resveratrol and PYK2 on Prostate Cells. International journal of molecular sciences 2016, 17, (9).

·         Conte, A.; Procaccini, C.; Iannelli, P.; Kisslinger, A.; De Amicis, F.; Pierantoni, G. M.; Mancini, F. P.; Matarese, G.; Tramontano, D., Effects of Resveratrol on P66shc Phosphorylation in Cultured Prostate Cells. Transl Med Unisa 2015, 13, 47-58.

In the new version, the reference 11 is  deleted.

At the line 83, the  reference 9 is deleted.

The reference 12 is deleted and replaced with the following new references:

·         Lecomte, S.; Lelong, M.; Bourgine, G.; Efstathiou, T.; Saligaut, C.; Pakdel, F., Assessment of the potential activity of major dietary compounds as selective estrogen receptor modulators in two distinct cell models for proliferation and differentiation. Toxicology and applied pharmacology 2017, 325, 61-70.

·         Chakraborty, S.; Kumar, A.; Butt, N. A.; Zhang, L. F.; Williams, R.; Rimando, A. M.; Biswas, P. K.; Levenson, A. S., Molecular insight into the differential anti-androgenic activity of resveratrol and its natural analogs: in silico approach to understand biological actions. Molecular bioSystems 2016, 12, (5), 1702-1709.

The reference 13 and 14 are deleted and replaced with the following new references:

·         Singhal, H.; Greene, M. E.; Tarulli, G.; Zarnke, A. L.; Bourgo, R. J.; Laine, M.; Chang, Y. F.; Ma, S. H.; Dembo, A. G.; Raj, G. V.; Hickey, T. E.; Tilley, W. D.; Greene, G. L., Genomic agonism and phenotypic antagonism between estrogen and progesterone receptors in breast cancer. Sci Adv 2016, 2, (6).

·         Maggi, A.; Villa, A., In vivo dynamics of estrogen receptor activity: The ERE-Luc model. Journal of Steroid Biochemistry and Molecular Biology 2014, 139, 262-269.

The reference 15 is deleted and replaced with the following new references:

·         Ferrero, G.; Miano, V.; Beccuti, M.; Balbo, G.; De Bortoli, M.; Cordero, F., Dissecting the genomic activity of a transcriptional regulator by the integrative analysis of omics data. Scientific reports 2017, 7.

·         Yi, P.; Wang, Z.; Feng, Q.; Chou, C. K.; Pintilie, G. D.; Shen, H.; Foulds, C. E.; Fan, G. Z.; Serysheva, I.; Ludtke, S. J.; Schmid, M. F.; Hung, M. C.; Chiu, W.; O'Malley, B. W., Structural and Functional Impacts of ER Coactivator Sequential Recruitment. Molecular cell 2017, 67, (5), 733-+.

The reference 17 is deleted.

The reference 18 is deleted and replaced with the following new reference:

·         Vanneste, M.; Hanoux, V.; Bouakka, M.; Bonnamy, P. J., Hyaluronate synthase-2 overexpression alters estrogen dependence and induces histone deacetylase inhibitor-like effects on ER-driven genes in MCF7 breast tumor cells. Molecular and cellular endocrinology 2017, 444, (C), 48-58.

The references 23 is deleted and replaced with the following new reference:

·         Merino, D.; Lok, S. W.; Visvader, J. E.; Lindeman, G. J., Targeting BCL-2 to enhance vulnerability to therapy in estrogen receptor-positive breast cancer. Oncogene 2016, 35, (15), 1877-1887.

The reference 28 is deleted and replaced with the following new reference:

·         Park, S. J.; Kim, J. G.; Kim, N. D.; Yang, K.; Shim, J. W.; Heo, K., Estradiol, TGF-1 and hypoxia promote breast cancer stemness and EMT-mediated breast cancer migration. Oncology letters 2016, 11, (3), 1895-1902.

The reference 29 is deleted and replaced with the following new reference:

·         Nagini, S., Breast Cancer: Current Molecular Therapeutic Targets and New Players. Anti-Cancer Agent Me 2017, 17, (2), 152-163.

At the line 141, the  reference 13 (Sommer S et al, 2001) and 15 (Schiff R et al, 2004) are deleted and replaced with the following new reference:

·         Lousberg, L.; Collignon, J.; Jerusalem, G., Resistance to therapy in estrogen receptor positive and human epidermal growth factor 2 positive breast cancers: progress with latest therapeutic strategies. Ther Adv Med Oncol 2016, 8, (6), 429-449.

The references 31 is deleted and replaced with the following new reference:

·         Baker, M. E.; Lathe, R., The promiscuous estrogen receptor: Evolution of physiological estrogens and response to phytochemicals and endocrine disruptors. Journal of Steroid Biochemistry and Molecular Biology 2018, 184, 29-37.

The reference 33 is deleted and replaced with the following new reference:

·         Saluzzo J, Hallman KM, Aleck K, Dwyer B, Quigley M, Mladenovik V, Siebert AE, Dinda S. The regulation of tumor suppressor protein, p53, and estrogen receptor (ERα) by resveratrol in breast cancer cells. Genes Cancer. 2016 Nov;7(11-12):414-425.

The references 40 is deleted and replaced with the following new reference:

·         Sinha, D.; Sarkar, N.; Biswas, J.; Bishayee, A., Resveratrol for breast cancer prevention and therapy: Preclinical evidence and molecular mechanisms. Semin Cancer Biol 2016, 40-41, 209-232.

The reference 41 is deleted and replaced with the following new reference:

·         Menazza, S.; Murphy, E., The Expanding Complexity of Estrogen Receptor Signaling in the Cardiovascular System. Circulation research 2016, 118, (6), 994-1007.

The reference 44 is deleted and replaced with the following new reference:

·         Coutinho, D. D.; Pacheco, M. T.; Frozza, R. L.; Bernardi, A., Anti-Inflammatory Effects of Resveratrol: Mechanistic Insights. International journal of molecular sciences 2018, 19, (6).

The reference 46 is deleted and replaced with the following new reference:

·         Rossi, E. L.; Khatib, S. A.; Doerstling, S. S.; Bowers, L. W.; Pruski, M.; Ford, N. A.; Glickman, R. D.; Niu, M. M.; Yang, P. Y.; Cui, Z. R.; DiGiovanni, J.; Hursting, S. D., Resveratrol inhibits obesity-associated adipose tissue dysfunction and tumor growth in a mouse model of postmenopausal claudin-low breast cancer. Molecular carcinogenesis 2018, 57, (3), 393-407.

At the line 250, the  reference 2 is deleted and replaced with the following reference:

·         Gucalp, A.; Traina, T. A., The Androgen Receptor: Is It a Promising Target? Ann Surg Oncol 2017, 24, (10), 2876-2880.

The references 47 is deleted and replaced with the following new reference:

·         Jernberg, E.; Bergh, A.; Wikstrom, P., Clinical relevance of androgen receptor alterations in prostate cancer. Endocr Connect 2017, 6, (8), R146-R161.

The reference 48 is deleted and replaced with the following new reference:

·         Wang, Q.; Wang, H.; Ju, Q.; Ding, Z.; Ge, X.; Shi, Q. M.; Zhou, J. L.; Zhou, X. L.; Zhang, J. P.; Zhang, M. R.; Yu, H. M.; Xu, L. C., The co-regulators SRC-1 and SMRT are involved in interleukin-6-induced androgen receptor activation. Eur Cytokine Netw 2016, 27, (4), 108-113.

The reference 49 is deleted and replaced with the following new reference:

·         Lasko, L. M.; Jakob, C. G.; Edalji, R. P.; Qiu, W.; Montgomery, D.; Digiammarino, E. L.; Hansen, T. M.; Risi, R. M.; Frey, R.; Manaves, V.; Shaw, B.; Algire, M.; Hessler, P.; Lam, L. T.; Uziel, T.; Faivre, E.; Ferguson, D.; Buchanan, F. G.; Martin, R. L.; Torrent, M.; Chiang, G. G.; Karukurichi, K.; Langston, J. W.; Weinert, B. T.; Choudhary, C.; de Vries, P.; Van Drie, J. H.; McElligott, D.; Kesicki, E.; Marmorstein, R.; Sun, C. H.; Cole, P. A.; Rosenberg, S. H.; Michaelides, M. R.; Lai, A.; Bromberg, K. D., Discovery of a selective catalytic p300/CBP inhibitor that targets lineage-specific tumours. Nature 2017, 550, (7674), 128.

The reference 50 is deleted and replaced with the following new reference:

·         Wadosky, K. M.; Koochekpour, S., Androgen receptor splice variants and prostate cancer: From bench to bedside. Oncotarget 2017, 8, (11), 18550-18576.

The reference 51 and 52 are deleted and replaced with the following new references:

·         McGuinness D, McEwan IJ. Posttranslational Modifications of Steroid Receptors: Phosphorylation. Methods Mol Biol. 2016;1443:105-17.

·         Deng, Q.; Zhang, Z.; Wu, Y.; Yu, W. Y.; Zhang, J. W.; Jiang, Z. M.; Zhang, Y.; Liang, H.; Gui, Y. T., Non-Genomic Action of Androgens is Mediated by Rapid Phosphorylation and Regulation of Androgen Receptor Trafficking. Cellular Physiology and Biochemistry 2017, 43, (1), 223-236.

The reference 53 is deleted and replaced with the following new reference:

·         Thoma C. Prostate cancer: Role for EGFR &HER2 in bone metastasis. Nat Rev Urol. 2017 Jan;14(1):7. doi: 10.1038/nrurol.2016.241

The reference 54 is deleted and replaced with the following new reference:

·         Chen, J. Q.; Li, L.; Yang, Z.; Luo, J.; Yeh, S. Y.; Chang, C. S., Androgen-deprivation therapy with enzalutamide enhances prostate cancer metastasis via decreasing the EPHB6 suppressor expression. Cancer letters 2017, 408, 155-163.

The references 55 and 56 are deleted and replaced with the following new references:

·         Galletti, G.; Leach, B. I.; Lam, L.; Tagawa, S. T., Mechanisms of resistance to systemic therapy in metastatic castration-resistant prostate cancer. Cancer treatment reviews 2017, 57, 16-27.

·         Le, B.; Powers, G. L.; Tam, Y. T.; Schumacher, N.; Malinowski, R. L.; Steinke, L.; Kwon, G.; Marker, P. C., Multi-drug loaded micelles delivering chemotherapy and targeted therapies directed against HSP90 and the PI3K/AKT/mTOR pathway in prostate cancer. PloS one 2017, 12, (3).

The references 58 and 59 are deleted and replaced with the following new references:

·         Brizzolara, A.; Benelli, R.; Vene, R.; Barboro, P.; Poggi, A.; Tosetti, F.; Ferrari, N., The ErbB family and androgen receptor signaling are targets of Celecoxib in prostate cancer. Cancer letters 2017, 400, 9-17.

·         Patek, S. C.; Willder, J. M.; Heng, J. S.; Taylor, B.; Horgan, P. G.; Leung, H. Y.; Underwood, M. A.; Edwards, J., Androgen receptor phosphorylation status at serine 578 predicts poor outcome in prostate cancer patients. Oncotarget 2017, 8, (3), 4875-4887.

The reference 60 is deleted and replaced with the following new reference:

·         Deng, Q.; Zhang, Z.; Wu, Y.; Yu, W. Y.; Zhang, J. W.; Jiang, Z. M.; Zhang, Y.; Liang, H.; Gui, Y. T., Non-Genomic Action of Androgens is Mediated by Rapid Phosphorylation and Regulation of Androgen Receptor Trafficking. Cellular Physiology and Biochemistry 2017, 43, (1), 223-236.

The reference 61 is deleted and replaced with the following new reference:

·         Kang, M.; Lee, K. H.; Lee, H. S.; Park, Y. H.; Jeong, C. W.; Ku, J. H.; Kim, H. H.; Kwak, C., PDLIM2 Suppression Efficiently Reduces Tumor Growth and Invasiveness of Human Castration-Resistant Prostate Cancer-Like Cells. The Prostate 2016, 76, (3), 273-285.

The reference 63 is deleted and replaced with the following new reference:

·         Gorowska-Wojtowicz, E.; Hejmej, A.; Kaminska, A.; Pardyak, L.; Kotula-Balak, M.; Dulinska-Litewka, J.; Laidler, P.; Bilinska, B., Anti-androgen 2-hydroxyflutamide modulates cadherin, catenin and androgen receptor phosphorylation in androgen-sensitive LNCaP and androgen-independent PC3 prostate cancer cell lines acting via PI3K/Akt and MAPK/ERK1/2 pathways. Toxicology in Vitro 2017, 40, 324-335.

·         Baron, S.; Manin, M.; Beaudoin, C.; Leotoing, L.; Communal, Y.; Veyssiere, G.; Morel, L., Androgen receptor mediates non-genomic activation of phosphatidylinositol 3-OH kinase in androgen-sensitive epithelial cells. Journal of Biological Chemistry 2004, 279, (15), 14579-14586.

The reference 64 is deleted and replaced with the following new reference:

·         Hamzeh, M.; Robaire, B., Androgens activate mitogen-activated protein kinase via epidermal growth factor receptor/insulin-like growth factor 1 receptor in the mouse PC-1 cell line. Journal of Endocrinology 2011, 209, (1), 55-64

The reference 65 is deleted and replaced with the following new reference:

·         Xia, D.; Lai, D. V.; Wu, W. J.; Webb, Z. D.; Yang, Q.; Zhao, L. C.; Yu, Z. X.; Thorpe, J. E.; Disch, B. C.; Ihnat, M. A.; Jayaraman, M.; Dhanasekaran, D. N.; Stratton, K. L.; Cookson, M. S.; Fung, K. M.; Lin, H. K., Transition from androgenic to neurosteroidal action of 5 alpha-androstane-3 alpha, 17 beta-diol through the type A gamma-aminobutyric acid receptor in prostate cancer progression. Journal of Steroid Biochemistry and Molecular Biology 2018, 178, 89-98.

The reference 67 is deleted and replaced with the following new reference:

·         Eiro, N.; Fernandez-Gomez, J.; Sacristan, R.; Fernandez-Garcia, B.; Lobo, B.; Gonzalez-Suarez, J.; Quintas, A.; Escaf, S.; Vizoso, F., Stromal factors involved in human prostate cancer development, progression and castration resistance. Journal of cancer research and clinical oncology 2017, 143, (2), 351-359.

The reference 68 is deleted and replaced with the following new reference:

·         Palethorpe, H. M.; Leach, D. A.; Need, E. F.; Drew, P. A.; Smith, E., Myofibroblast androgen receptor expression determines cell survival in co-cultures of myofibroblasts and prostate cancer cells in vitro. Oncotarget 2018, 9, (27), 19100-19114.

The reference 70 is deleted and replaced with the following new reference:

·         Goreczny, G. J.; Forsythe, I. J.; Turner, C. E., Hic-5 regulates fibrillar adhesion formation to control tumor extracellular matrix remodeling through interaction with tensin1. Oncogene 2018, 37, (13), 1699-1713.

The reference 72 is deleted and replaced with the following new reference:

·         Al Aameri, R. F. H.; Sheth, S.; Alanisi, E. M. A.; Borse, V.; Mukherjea, D.; Rybak, L. P.; Ramkumar, V., Tonic suppression of PCAT29 by the IL-6 signaling pathway in prostate cancer: Reversal by resveratrol. PloS one 2017, 12, (5).

2. Line 149-152 “Specifically, RSV may act as a super-agonist at moderate concentrations (10-25 mM) activating hormone receptor mediated gene transcription. At low concentrations (0.1-1 mM) RSV acts as anti-estrogen triggering pathways that inhibit estrogen dependent effects involved in proliferation and transformation.” Need to add with references.

Sorry for the error present in the text (mM instead of μM). The correct period is the following:

“Specifically, RSV may act as a super-agonist at moderate concentrations (10-25 μM) activating hormone receptor mediated gene transcription. At low concentrations (0.1-1 μM) RSV acts as anti-estrogen triggering pathways that inhibit estrogen dependent effects involved in proliferation and transformation (Basly JP, Marre-Fournier F, Le Bail JC, Habrioux G, Chulia AJ. Estrogenic/antiestrogenic and scavenging properties of (E)- and (Z)-resveratrol. Life Sci. 2000 Jan 21;66(9):769-77)

3.Line 173 “Several studies prove…” this sentence should follow with more than one reference.

As requested by the reviewer, in the new version, at the line 177, we added the following references:

·         Fernandes, G. F. S.; Silva, G. D. B.; Pavan, A. R.; Chiba, D. E.; Chin, C. M.; Dos Santos, J. L., Epigenetic Regulatory Mechanisms Induced by Resveratrol. Nutrients 2017, 9, (11).

·         Lee, P. S.; Chiou, Y. S.; Ho, C. T.; Pan, M. H., Chemoprevention by resveratrol and pterostilbene: Targeting on epigenetic regulation. Biofactors  2018, 44, (1), 26-35.

4.Please revise the format of every “et al” as “et al.”

In the new version, we revised the format as requested.

5.Line 310 “dependent manner-” should be “dependent manner.”.

In the new version, we made the correction.